# Establishing a Multidisciplinary Team-Based Pleural Service in the Era of Financial Austerity: The Role of the Thoracic Surgeon

**DOI:** 10.3390/medicina59030432

**Published:** 2023-02-22

**Authors:** Emmanouil I. Kapetanakis, Tatiana Sidiropoulou, Ioannis P. Tomos, Christos F. Kampolis, Thomas Raptakis, Periklis I. Tomos

**Affiliations:** 1Department of Thoracic Surgery, “Attikon” University Hospital, National and Kapodistrian University of Athens, 12462 Athens, Greece; 22nd Department of Anesthesiology, “Attikon” University Hospital, National and Kapodistrian University of Athens, 12462 Athens, Greece; 32nd Pulmonary Medicine Department, “Attikon” University Hospital, National and Kapodistrian University of Athens, 12462 Athens, Greece; 4Department of Clinical Pathophysiology, General Hospital of Athens “Laiko”, National and Kapodistrian University of Athens, 11527 Athens, Greece

**Keywords:** medical thoracoscopy-pleuroscopy, video assisted thoracoscopy, pleural disease, pleural service, financial crisis, austerity

## Abstract

Medical thoracoscopy/pleuroscopy has become, after bronchoscopy, the second most commonly utilized endoscopic procedure in interventional pulmonology. Due to their common origin, medical thoracoscopy/pleuroscopy and video-assisted thoracic surgery (VATS) are quite similar procedures technically. In contrast to the prevailing attitude that it should predominantly be performed by interventional pulmonologists, we believe that, like all hybrid-in-nature techniques, it should be implemented as part of a combined specialist care service/team. Herewith, we describe our attempt to establish a multidisciplinary pleural disease program during a difficult economic period for our country, comprising thoracic surgeons, pulmonologists and anesthesiologists, all of whom brought in their experience, expertise and resources to establish and develop the service resulting in a hybridization of the technique, with, as reported, quite favorable results.

Medical thoracoscopy/pleuroscopy has been increasingly performed nowadays by pulmonologists and is the second most common endoscopic procedure after bronchoscopy in interventional pulmonology, being integral for a specialist pleural disease service [1,2,3,4]. Its prevalence seems to be gradually growing, especially following the establishment of clear training pathways and society-sponsored courses [4,5]. In a recently conducted survey in the United Kingdom, centers with dedicated medical thoracoscopy/pleuroscopy services available had increased from 17 to 50, while Froudarakis et al., in another recent survey, demonstrated that following a European Respiratory Society (ERS)-sponsored medical thoracoscopy course, 58.6% of responders had either created or modified a medical thoracosopy/pleuroscopy program in their workplace [4,5].

Its advocates acclaim its minimally invasive nature, performing it as a day-care procedure in endoscopic suites using conscious sedation, compared to video-assisted thoracic surgery (VATS), conducted in operating theaters under general anesthesia by surgical teams, which usually requires hospitalization afterwards [6]. The technique’s simplicity, the relatively minor logistics required and its favorable safety record are key factors for its proliferation over recent years [3,6].

Given their common origin, medical thoracoscopy/pleuroscopy and VATS are quite similar procedures, and truthfully, the boundary line between the two techniques is quite blurry [7]. According to the literature, a number of surgeons, including authors, have performed VATS in awake, non-intubated patients under local anesthesia and sedation [7]. Similarly, advances in optics, video technology and instrumentation have allowed the performing of medical thoracoscopy/pleuroscopy using video assistance, thus making it analogous to VATS [7].

In contrast to the prevailing attitude that medical thoracoscopy/pleuroscopy should be performed by interventional pulmonologists exclusively, we believe that, like all hybrid techniques, it should be part of a combined specialist care service [3,7].

The current financial austerity experienced by Greece since 2008 has significantly affected the extent and quality of the medical services provided by the Greek National Healthcare System (GrNHS) [8,9,10]. Applied fiscal-control-centered policies have augmented problems such as understaffing and aging of the healthcare workforce, arduous shift patterns and staff fatigue, reductions in personnel compensation, and shortages of medical and surgical supplies [8,9,10,11].

In these challenging times, it seemed impossible to keep existing, let alone create and establish new healthcare provision structures [8,9,10]. However, wishing to provide comprehensive pleural disease management not only regionally but countrywide (our university hospital being a centrally located tertiary care facility acting as a referral center for patients from all over Greece), a pleural service performing medical thoracoscopy/pleuroscopy was established in January 2018, organized and managed by the senior thoracic surgeon, his senior specialty trainee (now a locum consultant), two pulmonologists and a dedicated thoracic anesthesiologist. A regular outpatient clinic for new and follow-up patients was established once weekly, and a theater list was allocated as required. 

During the first year of operation, 50 patients underwent medical thoracoscopy/pleuroscopy to diagnose/manage their pleural effusion by this combined team of thoracic surgeons, pulmonologists and anesthesiologists. Patient demographics and indications for medical thoracoscopy/pleuroscopy are presented in Table 1. Most patients had left-sided pleural effusions, mostly non-infective exudates (Table 1). The principal indication for medical thoracoscopy/pleuroscopy was an undiagnosed pleural effusion followed by refractory malignant pleural effusions (Table 1). Most procedures were diagnostic or combined diagnostic and therapeutic using talc poudrage pleurodesis (Table 1). A small number of patients with infected parapneumonic effusions underwent drainage with the break-up of loculations and adhesions to facilitate lung re-expansion (Table 1).

Unsurprisingly, malignancy, either non-small cell lung cancer, metastatic diseases of non-thoracic primary origins or mesothelioma were the principal diagnoses obtained (Table 1). Only in one case did medical thoracoscopy/pleuroscopy fail to produce a diagnosis, producing an overall diagnostic yield of 98% (Table 1). Most complications observed were minor in nature, considering the associated co-morbidities of the patients, and there were no conversions to general anesthesia (Table 1). The incidence of complications was slightly higher in certain variables than those reported in the literature, but this can be attributed to the associated learning curve and the small sample of initially reported patients [12,13].

Naturally, forming a new pleural disease service in this era of austerity is challenging. A number of obstacles needed to be overcome, both administrative and financial in nature. The financial constraints imposed by the crisis meant that our team often had to deal with shortages in supplies and materials. We became quite inventive (a 10 cc syringe with its tip cut off makes an excellent chest trocar, and a bronchoscopy biopsy forceps can be easily re-appropriated to be used via the Olympus LFT-160 semi-rigid thoracoscope) but also organized and proactive in terms of ordering materials in advance (such as talc, which needed a long procurement approval process despite its low cost because it was classified neither as “medication” nor as “surgical consumables”).

Consequently, in our hospital, the pleural disease service program is unique because of its multidisciplinary team approach promoting concurrent active participation by thoracic surgeons, pulmonologists and thoracic anesthetists. Patients are referred to the pleural disease service either by the respiratory or internal medicine teams or by the thoracic service, both internally and from regional hospitals. Following a discussion where the existence of previously established indications (as per guidelines) is confirmed, and the appropriateness for intervention is decided, the patient would undergo medical thoracoscopy/pleuroscopy by this combined team of thoracic surgeons, pulmonologists and anesthetists [14]. Initially, procedures were performed by an experienced thoracic surgeon and an operator in training (either a surgeon or a pulmonologist), but with time and familiarity/competence, senior surgical oversight was scaled back. However, a surgeon would always participate in the procedure either as the primary operator or as an assistant. Patients were co-followed up in their admitting ward by the pleural and the referring teams until their indwelling chest drain was removed. Following this, the care and discharge of the patient was decided by the admitting team. The co-performance of medical thoracoscopy/pleuroscopy procedures jointly by surgeons and pulmonologists under anesthetic support offers numerous appreciable benefits. In our case, all three specialties brought in their experience, expertise and resources to establish and develop the service resulting in a technique hybridization, with, as reported herewith, quite favorable results. Determination, flexibility, teamwork and a lot of industriousness from all its members were instrumental in the success of the service.

The role of a thoracic surgeon in such a team is significant. Specifically, a surgeon’s perspective and knowledge of surgically amiable pathologies and risk assessment mean that during multidisciplinary team (MDT) discussions, patients are more accurately evaluated and appropriately allocated to either medical thoracoscopy/pleuroscopy or VATS procedures. Our dedicated thoracic anesthetists’ role is also paramount in this, assessing fitness for and mode of anesthesia.

Experienced surgeons are accustomed to appraising and identifying intra-thoracic pathologies, having observed them from early career stages. Therefore, intra-operatively they can more accurately recognize various pathologies and correspondingly help the team manage them.

Moreover, the surgeons’ familiarity with thoracic topographical anatomy can assist during pre-procedural planning, selecting entry points more precisely and proximal to desired areas of inspection/biopsy while simultaneously avoiding high-risk regions. Obviously, surgeons are quite capable of performing biopsies safely, being more aware of the location, identification and course of vessels and nerves. In addition, surgeons are more accustomed to hemorrhage, recognizing its severity more easily, thus responding and managing it better. Naturally, by their simple participation, they provide surgical back-up during emergencies.

Surgeons, through their training, gain familiarity with the visual anatomy produced by laparoscopy/thoracoscopy, becoming accustomed to the associated video equipment and surgical instruments. Moreover, their acquired training and enhanced manual dexterity allow them to manipulate the thoracoscope and the associated surgical tools more competently, thus possibly opening a role in instructing the other team members in their usage. Our team mainly utilizes the Olympus LFT-160 semi-rigid thoracoscope, which because of its shared appearance and functionality to a bronchoscope, makes it more familiar for our pulmonologist to use, but rigid thoracoscopy is also often used.

Therefore, we feel that trainee pulmonologists who wish to pursue interventionist careers should, from an early period in their training, rotate through a thoracic surgery service or participate in MDT pleural services such as this, to gain experience and self-confidence. This will help them become more independent and will also promote cross-collaboration with thoracic surgeons and anesthetists.

The benefits of a combined multidisciplinary team of surgeons, pulmonologists and anesthetists running a pleural disease service together are supported by the reported outcomes, especially the high (98%) diagnostic yield achieved with medical thoracoscopy/pleuroscopy and the low complication incidence. This bridging of specialties allows categorizing patients with pleural disease better and more appropriately directing them for VATS when, for example, extensive adhesions, trapped lung or increased bleeding diathesis is suspected, or for medical thoracoscopy/pleuroscopy, if frailer with poorer performance status and/or when general anesthesia is not tolerated. Naturally, there may be some criticism of cost increases by this combined team approach, but we believe that the benefits in the complexity of cases managed and also in patient safety and efficacy conferred by the participation of surgeons and anesthetists more than justify it. Furthermore, although a comprehensive cost analysis is not possible due to the per diem reimbursement model employed by GrNHS hospitals, the diagnosis-related group (DRG) payment code for medical thoracoscopy versus VATS is €200.0 less, which, although small, actually represents a cost reduction and net savings.

In conclusion, we believe that thoracic surgeons can play a significant directional and educational role in a medical thoracoscopy/pleuroscopy service, and thus an attitude of closer collaboration between specialties should be fostered and cultivated.

## Figures and Tables

**Table 1 medicina-59-00432-t001:** Patient characteristics, procedures, diagnosis and complications.

	(*n* = 50)

Age *(mean age in years)*	62.9
Male gender, *(n,%)*	26 (52%)
Median American Society of Anesthesiologists (ASA) Physical Classification Score	3
Mean Pre-Operative PaO_2_/FiO_2_ Ratio	284.1

** *Indications for Thoracoscopy (n,%):* **	
Right-sided effusion	21 (42%)
Left-sided effusion	29 (58%)
Pleural effusion of unknown origin	37 (74%)
Known malignant pleural effusion and failure of conventional talc pleurodesis	9 (18%)
Recurrent idiopathic pleural effusion *(Nephritic syndrome)*	1 (2%)
Parapneumonic effusion/empyema	3 (6%)

** *Procedures performed (n,%):* **	
Pleural drainage and biopsy	21 (42%)
Combined drainage, biopsy and pleurodesis	17 (34%)
Pleural drainage and pleurodesis	9 (18%)
Drainage of empyema	3 (6%)

** *Diagnosis (n,%):* **	
Non-small cell lung cancer	5 (10%)
Mesothelioma	6 (12%)
Metastatic non-thoracic origin disease	12 (24%)
Non-Hodgkin’s lymphoma	3 (6%)
Tuberculosis	1 (2%)
Chronic inflammation	2 (4%)
Systemic Lupus Erythematosus	4 (8%)
Nephritic syndrome	1 (2%)
Negative biopsy for malignancy	3 (6%)
Non-diagnostic biopsy	1 (2%)

** *Complications (n,%):* **	
Subcutaneous emphysema	7 (14%)
Minor bleeding	3 (6%)
Fever and intravenous antibiotic therapy	3 (6%)
Failure of pleurodesis—discharged with drain in situ	1 (2%)

## Data Availability

Data available upon prompt request from the pleural disease service research database.

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
