# Peer review of "Establishing a Multidisciplinary Team-Based Pleural Service in the Era of Financial Austerity: The Role of the Thoracic Surgeon"

_medicina, 2023, doi:10.3390/medicina59030432_

Round 1
Reviewer 1 Report (Previous Reviewer 1)
dear authors
congrats on the good job
Reviewer 2 Report (Previous Reviewer 2)
.
This manuscript is a resubmission of an earlier submission. The following is a list of the peer review reports and author responses from that submission.
Round 1
Reviewer 1 Report
I congratulate the authors for the good idea behind this communication. However, I have some curiosities and suggestions. I think that the power of the message can be improved by providing a short comparison of the costs and outcomes of this service compared to OR thoracoscopy, to better justify the need for such service during a financial crisis. I wouldn’t be surprised if the presence of this service would result in an overall cost reduction due to the reduction of OR thoracoscopies. Moreover, to be more attractive for general thoracic surgeons worldwide I would provide a brief overview of the shortages the thoracic surgeon colleagues faced in Greece during the financial crisis.
Reviewer 2 Report
Dear authors,
thank you for submitting as a communication your project for a multidisciplinary team-based pleural service and the important rule of the thoracic surgeon.
But unfortunatly there are a lot of problems that needs to be clarified.
First you stated that medical thoracoscopy is been increasingly performed. The references for this statement are from 2001 ans 2011. This is not the newest an by far not the developement in other european countries. So can you please clarify your statement with precise data and references.
A flow chat of the decision making . when is a medical thoracoscoy be done and when should it be done or is indicated by a thorcic surgeon will help to understand the way your team ist working. What is the precise rule of the surgeon. Is he doing the procedure. who is lookong after the patient in the hospital on the ward? Time of discharge?
What was the coversion rate from medical to surgical procedure. How many chest drains are placed , woh removed them?
The complication rate seems high, can you explan ist or compare it with literature and expecially surgical or mediacal procedures without the team structure.
What is the curiculum for the peumologist? When is he allowed to do this procedure on is own? How many procedured are needed?
And at least: what is the advantage for a medical thoracoscopy when the surgeon is more familiar with the handling of the complications.
As you are publishing in an international ( english writen) journal the special greece perspective should be clearly stated and be discussed critically in an least a european context.